# ATHENA: ENHANCING MULTIMODAL REASONING WITH DATA-EFFICIENT PROCESS REWARD MODELS

## ABSTRACT

We present Athena-PRM, a multimodal process reward model (PRM) designed to evaluate the reward score for each step in solving complex reasoning problems. Developing high-performance PRMs typically demands significant time and financial investment, primarily due to the necessity for step-level annotations of reasoning steps. Conventional automated labeling methods, such as Monte Carlo estimation, often produce noisy labels and incur substantial computational costs. To efficiently generate high-quality process-labeled data, we propose leveraging prediction consistency between weak and strong completers as a criterion for identifying reliable process labels. Remarkably, Athena-PRM demonstrates outstanding effectiveness across various scenarios and benchmarks with just 5,000 samples. Furthermore, we also develop two effective strategies to improve the performance of PRMs: ORM initialization and up-sampling for negative data. We validate our approach in three specific scenarios: verification for test-time scaling, direct evaluation of reasoning step correctness, and reward ranked fine-tuning. Our Athena-PRM consistently achieves superior performance across multiple benchmarks and scenarios. Notably, when using Qwen2.5-VL-7B as the policy model, Athena-PRM enhances performance by 10.2 points on WeMath and 7.1 points on MathVista for test-time scaling. Furthermore, Athena-PRM sets the state-of-the-art (SoTA) results in VisualProcessBench and outperforms the previous SoTA by 3.9 F1-score, showcasing its robust capability to accurately assess the correctness of the reasoning step. Additionally, utilizing Athena-PRM as the reward model, we develop Athena-7B with reward ranked fine-tuning and outperforms baseline with a significant margin on five benchmarks.

## 1 INTRODUCTION

In recent years, Large Language Models (LLMs) (Brown et al., 2020; Anthropic, 2025; Touvron et al., 2023; Liu et al., 2024; Yang et al., 2024) have achieved remarkable success in natural language processing. Building on this, Multimodal Large Language Models (MLLMs) (Liu et al., 2023; Li et al., 2025; Bai et al., 2025; Chen et al., 2024) have made significant strides in various vision-language tasks, such as visual question answering (Antol et al., 2015; Mathew et al., 2021) and chart understanding (Masry et al., 2022). Despite their impressive performance, solving complex tasks involving mathematical and multi-step reasoning remains challenging.

To enhance reasoning capabilities, several approaches have been explored, including fine-tuning on long chain-of-thought (CoT) data (Wei et al., 2022; Muennighoff et al., 2025), offline preference optimization (Wang et al., 2024c; Pang et al., 2024), and online reinforcement learning (Shao et al., 2024; Guo et al., 2025; Hu, 2025; Team et al., 2025). Another promising avenue is test-time scaling (TTS), which involves generating multiple responses from a policy model and selecting the most consistent answer (Wang et al., 2023) or the solution with the highest reward using reward models (Cobbe et al., 2021; Lightman et al., 2024; Uesato et al., 2022; Snell et al., 2025; Setlur et al., 2025; Wang et al., 2024b).

Reward models for TTS mainly include two types: outcome reward models (ORMs) (Cobbe et al., 2021) and process reward models (PRMs) (Lightman et al., 2024). ORMs evaluate the reward score for a given question and its solution, whereas PRMs provide reward scores for each intermediate reasoning step, offering fine-grained feedback. PRMs typically deliver superior performance and

**Question**
The difference between two perfect squares is 133. What is the smallest possible sum of the two perfect squares?

**Previous solution steps**
Let the two perfect squares be $x^2$ and $y^2$ where $x > y$. We are given that $x^2 - y^2 = 133$. We can factor the difference of squares as $(x + y)(x - y) = 133$. **Since $x + y > x - y$, we have that $x + y$ is a larger factor of 133. The prime factorization of 133 is $13 \times 10$ , so $x + y = 13$ and $x - y = 10$.**

**Weak completer**
Solving for $x$, we get $x = \frac{13+10}{2} = \frac{23}{2}$. Solving for $y$, we get $y = \frac{13-10}{2} = \frac{3}{2}$. Therefore, the smallest possible sum of the two perfect squares is

$$x^2 + y^2 = (\frac{23}{2})^2 + (\frac{3}{2})^2 = \frac{529}{4}$$

**Strong completer**
We can not have $x + y = 13$ and $x - y = 10$ because $x$ and $y$ would not be integers in that case. Solving the system of equations $x + y = 13$ and $x - y = 10$, we add the two equations to get $x = 11.5$. Since $x$ must be an integer, we need to check if there are other factor pairs of 133 that could work. The possible factor pairs of 133 are:
$x + y = 133, x - y = 1$ and $x + y = 19, x - y = 7$
Solve the two possible systems of equations.
(..........)
The final answer is 205.

Figure 1: Illustration of different completers under the same question and solution steps. Even if given wrong intermediate steps, the strong completer stills reach the final answer while the weak completer fails. We omitted some intermediate steps in the figure for simplicity.

better out-of-distribution generalization (Lightman et al., 2024). However, obtaining high-quality data with process labels poses significant challenges. PRM800K (Lightman et al., 2024) involves collecting 800K labeled steps through human annotations, which is time-consuming and requires skilled annotators, particularly for complex multi-step reasoning and mathematical tasks. Math-Shepherd (Wang et al., 2024b) proposes an automated labeling method using Monte Carlo (MC) estimation. It defines the quality of an intermediate step as its potential to reach the correct answer. MC estimation for step-level labels is typically performed by sampling numerous reasoning trajectories using a language model, referred to as the *completer*, to estimate the probability of arriving at the correct answer. However, this approach involves significant computational overhead. Besides, another shortcoming of MC-based estimation methods is that estimated step-level labels are inevitably noisy.

We aim to address the above challenges: reducing computational costs and mitigating noisy step labels. We first find that the accuracy of step labels estimated by MC-based methods is influenced by the reasoning capability of the completer. A strong completer can arrive at the correct answer despite incorrect intermediate steps as shown in Figure 1, whereas a weak completer may struggle even with correct intermediate steps. The estimation of MC-based methods is biased toward the completers we use. However, the correctness of the intermediate step should not depend on the completer. Based on this insight, **we propose using both weak and strong completers to estimate step labels, retaining only those steps where labels generated by both completers are consistent to remove the bias caused by completers**. This approach improves the quality of the step label. Empirical results demonstrate that a small set of high-quality labels ($\sim$5K) achieves impressive performance compared to large-scale data ($\sim$ 300K) labeled by vanilla MC-based estimation (Wang et al., 2024b). In addition, our approach requires only **1/45** of the GPU hours for data synthesis and **1/60** of the GPU hours for reward model training, significantly lowering computational costs.

After acquiring high-quality datasets, we explore two effective strategies for training PRMs: **initialization from ORMs** and **up-sampling data with negative labels**. PRMs are typically fine-tuned from pre-trained foundation models, such as LLaMA-3.1-8B-Instruct (Grattafiori et al., 2024) in RLHF-workflow (Dong et al., 2024). Previous studies indicate that ORMs possess some ability to assess intermediate step correctness through confidence variation (Lu et al., 2024a) or parameterization (Cui et al., 2025; Yuan et al., 2025). Inspired by this, we find that initializing PRMs from ORMs significantly boosts performance, as ORMs trained on large-scale *sample-level* data serve as pre-training with coarse-grained data. And PRMs can be considered fine-tuning on high-quality, fine-grained *step-level* data. Additionally, we show that *label imbalance* exists in process-labeled solutions and we propose to up-sample data with negative step labels to address this.

Building on these methodologies, we develop our outcome reward model Athena-ORM and process reward model Athena-PRM. Leveraging Athena-PRM, we introduce Athena-7B with reward ranked fine-tuning. We validate our approach across three scenarios: 1) test-time scaling (Snell et al., 2025): where Athena-PRM ranks multiple outputs generated by policies under the Best-of-N evaluation;

2) direct evaluation of the correctness of reasoning steps using Athena-PRM; 3) reward ranked fine-tuning (Dong et al., 2023): where Athena-PRM ranks outputs sampled from the current policy, utilizing the highest reward response for policy model fine-tuning. In the TTS scenario, we evaluate Athena-PRM on seven multimodal math and reasoning benchmarks with three different policy models ranging from 7B to 72B, demonstrating significant improvements in reasoning abilities. For instance, on the WeMath benchmark (Qiao et al., 2025), Athena-PRM enhances the zero-shot baseline by **10.2** points using Qwen2.5-VL-7B (Bai et al., 2025) as the policy model. Furthermore, Athena-PRM excels in text-only benchmarks, achieving an **8.9** points improvement on a challenging math benchmark (Hendrycks et al., 2021) with Mistral-8B. To assess its ability to judge intermediate reasoning step correctness, Athena-PRM is evaluated on the VisualProcessBench (Wang et al., 2025), showcasing strong performance and outperforming VisualPRM-8B (Wang et al., 2025), an open-source multimodal PRM, and proprietary models as judge. In the reward ranked fine-tuning scenario, our fine-tuned model Athena-7B, based on Qwen2.5-VL-7B (Bai et al., 2025), significantly enhances the policy model's reasoning capabilities across seven math and reasoning benchmarks.

Our contributions are summarized as follows:

- We propose using prediction consistency between weak and strong completers to filter noisy process labels, enhancing the quality of the automated-labeled process data. The high-quality process labeled data shows surprising data efficiency for training PRMs.

- We introduce two effective strategies to improve the performance of PRMs: ORM initialization and negative data up-sampling.

- We develop our reward models Athena-ORM and Athena-PRM, and evaluate them under the Best-of-N setting on nine math and reasoning benchmarks across various model sizes and families, demonstrating the effectiveness of our approach. Moreover, Athena-PRM achieves state-of-the-art (SoTA) performance in assessing the correctness of intermediate steps directly on the VisualProcessBench (Wang et al., 2025).

- Leveraging Athena-PRM, we introduce Athena-7B, a MLLM with exceptional capabilities fine-tuned using reward ranked fine-tuning. Extensive experiments highlight the superiority of Athena-7B across diverse benchmarks.

## 2 METHOD

In this section, we first introduce basic concepts, including ORMs and PRMs in Sec. 2.1. Next, we present our method for constructing a high-quality PRM training set, along with practical training strategies to improve PRM performance in Sec. 2.2 and 2.3. The data curation process for training is detailed in Sec. 2.4. Finally, we discuss three application scenarios of PRMs in Sec. 2.5.

### 2.1 PRELIMINARIES:REWARD MODELS FOR MATHEMATICAL PROBLEM

**ORMs.** Given a mathematical problem $x$ with golden answer $y^*$ and its solution $a$ generated by policy $\pi$, i.e., $a \sim \pi(\cdot \mid x)$, ORMs assign a reward $r(x, a) \in (0, 1)$ to reflect the correctness of the solution $a$. To train ORMs, we take inputs as $(x, a)$ and predict correctness $\delta(y, y^*)$ of prediction with following loss function:

$$\mathcal{L}_{ORM} = -\left(\delta(y, y^*) \cdot \log r(x, a) + (1 - \delta(y, y^*)) \cdot \log(1 - r(x, a))\right), \quad (1)$$

where $\delta(y, y^*) = 1$ if $y = y^*$, otherwise $\delta(y, y^*) = 0$.

**PRMs.** Different from ORMs, PRMs aim to predict the correctness of *each step* in the solution. Specifically, we sample response $a \sim \pi(\cdot \mid x)$. A response $a$ usually consists of multiple reasoning steps separated by a delimiter (e.g. \n\n), a.k.a $a = (a_1, a_2, \ldots, a_K)$ when $K$ is the number of reasoning steps. To predict the correctness of each step, we train PRMs with the cross-entropy loss for each step in the solution. Formally, we get the following loss function:

$$\mathcal{L}_{PRM} = -\left(\sum_{i=1}^{K} \delta_i \cdot \log r(x, a_i) + (1 - \delta_i) \cdot \log(1 - r(x, a_i))\right), \quad (2)$$

where $\delta_i = 1$ if the step $a_i$ is correct, otherwise $\delta_i = 0$.

## 2.2 Automated Process Label Annotation

Compared with ORMs, PRMs usually achieve better performance and exhibit stronger out-of-distribution generalization (Lightman et al., 2024). However, collecting high-quality data with process labels is challenging. Human annotation provides accurate supervision signals (Lightman et al., 2024), but it requires expensive human labeling and difficult to scale up. Math-Shepherd (Wang et al., 2024b) proposes to use a Monte Carlo (MC) sampling method to estimate the correctness of each step without human supervision. Specifically, to estimate the correctness of step $a_i$, we use a *completer* $\phi$ to finalize the reasoning process from step $a_i$, and get the final answer $y$. We repeat sampling $T$ times and get corresponding answers $\mathcal{Y} = \{y_j\}_{j=1}^T$. There are two ways to estimate the correctness of $a_i$: soft estimation and hard estimation.

For soft estimation, we assume that the frequency of getting the correct answer could dictate the quality of a step:

$$\delta_i^{soft} = \frac{\sum_{j=1}^T \mathbb{I}(y_j = y^*)}{T}. \tag{3}$$

For hard estimation, we assume that the correct answer is right when the step could reach the answer in $T$ samples:

$$\delta_i^{hard} = \begin{cases} 1 & \text{if } \exists y_j \in \mathcal{Y}, y_j = y^*, \\ 0 & \text{otherwise.} \end{cases} \tag{4}$$

In this paper, we mainly discuss the hard estimation for PRMs because it allows us to use standard language modeling loss to train PRMs and eliminates the adjustments of the training pipeline.

MC-based estimation methods provide an automated and scalable labeling strategy for intermediate reasoning steps. However, automatically labeled data often contain incorrect labels, and MC-based methods typically incur substantial computational costs due to extensive sampling requirements. For instance, when processing a solution $a$ with $K$ steps, the *completer* $\phi$ must generate $T \times K$ solutions, which produces $T \times K$ times computation cost compared with ORMs.

To enhance the accuracy of the labeling process and reduce the computational burden of data synthesis, our central approach is to utilize a smaller set of high-quality data with process labels. We improve the quality of automatically labeled process labels by introducing consistency filtering between weak and strong completers. The rationale is straightforward: the results of hard estimation depend on the number of samples $T$, the difficulty of the problem $x$, and the ability of the completer $\phi$. A strong completer can arrive at the correct answer even when provided with incorrect steps, as shown in Figure 1, whereas a weak completer struggles to find the final answer even when given correct steps. We aim to enhance the quality of process labels through prediction consistency between weak and strong completers. Specifically, we only use steps whose assigned labels are consistent between different completers.

To validate our method, we conducted a simple experiment using the PRM800K dataset (Lightman et al., 2024). We employed Mistral-7B-Instruct (Jiang et al., 2023) as the weak completer $\phi_w$ and Qwen2.5-72B-Instruct (Yang et al., 2024) as the strong completer $\phi_s$, setting the number of samples $T = 8$ for MC-based estimation. We randomly sampled 50 queries and reported the accuracy of estimated process labels:

|  | Weak Completer $\phi_w$ | Strong Completer $\phi_s$ | Consistency Filter |
|---|---|---|---|
| Accuracy (%) | 78.2 | 83.4 | **94.1** |

Our experiments show that **label quality improves significantly after applying the consistency filter**. Empirically, we demonstrate that **a small number of high-quality labels is sufficient to achieve superior performance**, thereby reducing computational costs compared to the vanilla MC-based estimation (Wang et al., 2024b). Results of more strategies and combination of completers are provided in Tabele 6 of Appendix C.1.

## 2.3 Additional Strategies for Training PRMs

We offer two strategies for enhancing the performance of PRMs: initialization from ORMs and negative data up-sampling. First, initializing PRMs with ORMs trained on large-scale sample-level annotated data improves performance. Previous work indicates that ORMs possess a certain

capacity to assess the correctness of intermediate steps (Cui et al., 2025; Lu et al., 2024a; Yuan et al., 2025). We empirically show that PRMs benefit from a few high-quality examples when initialized with ORMs. Large-scale sample-level annotation data provides weaker supervision, with outcome supervision acting as a simplified form of process supervision. Training ORMs on large-scale sample-level annotations serves as a "pre-training", while training PRMs from pre-trained ORMs acts as "fine-tuning" with high-quality process-level annotated data.

Secondly, our findings reveal that *label imbalance* is prevalent in most process-labeled datasets, such as PRM800K (Lightman et al., 2024) and our synthetic data, where correct steps are more common than errors. We provide a detailed distribution of process labels in Table 9. We up-sample data with negative labels to tackle this problem and empirical results show that up-sampling data with negative labels improves performance with minimal additional computation.

## 2.4 TRAINING DATA CURATION

To construct a diverse and high-quality dataset, we first gather data from various public multimodal and text-only datasets. For multimodal datasets, we collect from MathV360k (Shi et al., 2024), UniGeo (Chen et al., 2022), Geometry3k (Lu et al., 2021), CLEVR-Math (Lindström & Abraham, 2022), ScienceQA (Lu et al., 2022), GeomVerse (Kazemi et al., 2023), GeoQA-plus (Cao & Xiao, 2022), and DocVQA (Mathew et al., 2021). For text-only datasets, we utilize GSM8K (Cobbe et al., 2021), MATH (Hendrycks et al., 2021), and NuminaMath-1.5 (LI et al., 2024).

If the original dataset includes different splits, we use only the training portion to prevent test set contamination. We remove all judgment/proof problems and convert multiple-choice questions to open-ended ones to deter policy models from "guessing" answers. Then, for each query, we sample 8 responses from the policy model. To ensure dataset diversity, we employ multiple models as policies, including InternVL2.5-8B/78B (Chen et al., 2024), Qwen2.5-VL-7/72B-Instruct (Bai et al., 2025), LLaVA-OneVision-7B/72B (Li et al., 2025) for multimodal datasets, and Qwen2.5-7/72B-Instruct (Yang et al., 2024) for text-only datasets.

Next, we apply carefully designed filtering rules to all responses. We exclude responses with too many or too few tokens and eliminate those exhibiting repetitive patterns. We employ $n$-gram deduplication to remove similar responses and enhance diversity. Ultimately, we obtain approximately 600K queries with corresponding responses. We parse answers from responses and assign correctness labels of 0 or 1 to each query-response pair. We refer to this dataset as Athena-600K and employ it to train our outcome reward model Athena-ORM.

For training PRMs, we generate process labels using the proposed method introduced in Sec. 2.2. We use Qwen2.5-VL-3B (Bai et al., 2025) as the weak completer $\phi_w$ and Qwen2.5-VL-72B (Bai et al., 2025) as the strong completer $\phi_s$. Finally, we collect approximately 5K samples for training our process reward model Athena-PRM. We also compare with PRMs trained using approximately 300K samples estimated by the vanilla MC-based method outlined in Wang et al. (2024b). We denote these datasets as Athena-5K and MC-300K, respectively, and compare their performance and computational cost in Sec. 3.4.

## 2.5 APPLICATION SCENARIOS

Following reward model training, we assess the performance of ORMs and PRMs in three scenarios: test-time scaling (Snell et al., 2025), direct judgment, and reward ranked fine-tuning (Dong et al., 2023).

**Verification for test-time scaling.** We adopt a Best-of-N evaluation paradigm. Specifically, given a problem $x$ in the test set, we sample $N$ solutions from the policy $\pi$. All solutions are scored using a reward model, and we choose the solution with the highest score. For ORMs, we directly use the outputs from ORMs as the reward of solutions. For PRMs, the minimum reward across all steps is used to rank all solutions.[1]

**Direct judgment for reasoning steps.** Besides verification at test-time under the Best-of-N setting, Athena-PRM can also be used to directly identify erroneous steps in the mathematical reasoning

---

[1]We test different choices as the reward for solutions in Appendix C.3.

process. Given a solution $a$ with $K$ steps as input, Athena-PRM outputs the correctness score $\{\delta_i\}_{i=1}^K$ for each step $\{a_i\}_{i=1}^K$ in a single forward pass.

**Response ranking for reward ranked fine-tuning.** We explore the use of PRMs for data synthesis in reward ranked fine-tuning (Dong et al., 2023; Yuan et al., 2023; Singh et al., 2024; Zeng et al., 2025). Using the current policy $\pi$, we generate $M = 8$ solutions for an input problem $x$. Subsequently, we filter out solutions with incorrect answers and apply deduplication to remove highly similar responses, enhancing the diversity of the remaining solutions. We retain queries where 2 to 6 out of 8 responses are correct, excluding those with too few or too many correct answers, as such cases are either too easy or too challenging for the current policy $\pi$. Following this filtering step, we use Athena-PRM to score all solutions for each query and select the solution with the highest reward as the corresponding label. The policy $\pi$ is fine-tuned using the synthetic data generated as described above.

## 3 EXPERIMENTS

To comprehensively evaluate the proposed method, we assess Athena-ORM and Athena-PRM under the Best-of-N setting on seven multimodal benchmarks and two text-only benchmarks in Sec. 3.1. Furthermore, we evaluate Athena-PRM on VisualProcessBench, comparing its performance with MLLMs as judges and a recent open-source multimodal PRM, VisualPRM-8B (Wang et al., 2025), as described in Sec. 3.2. We assess the capability of Athena-7B on multimodal math and reasoning tasks in Sec. 3.3. To validate the effectiveness of our designs for Athena-PRM, an in-depth analysis is provided in Sec. 3.4.

### 3.1 BEST-OF-N EVALUATION

**Benchmarks.** We evaluate Athena-ORM and Athena-PRM across multiple multimodal mathematical and logical reasoning benchmarks, including MathVista (Lu et al., 2024b), MathVision (Wang et al., 2024a), MathVerse (Zhang et al., 2024), WeMath (Qiao et al., 2025), DynaMath (Zou et al., 2025), LogicVista (Xiao et al., 2024), and MMMU (Yue et al., 2024). To validate the effectiveness of Athena-ORM and Athena-PRM in the text-only scenario, we conduct experiments on the GSM8K (Cobbe et al., 2021) and MATH (Hendrycks et al., 2021) under the BoN settings. Further details can be found in Appendix C.2.

**Settings.** We employ Athena-ORM and Athena-PRM as the reward model for BoN evaluation, sampling $N = 8$ solutions for every problem by default. For decoding, we set the temperature to 0.8 and use nucleus sampling (Holtzman et al., 2020) with top-$p$ set to 0.9, and report the average of five runtimes. For multimodal benchmarks, we choose Qwen2.5-VL-7B/72B (Bai et al., 2025) and InternVL2.5-8B (Chen et al., 2024) as policy models.[2] For text-only datasets, we choose Qwen2.5-7/72B (Yang et al., 2024) and Ministral-8B[3] as policy models. Zero-shot and self-consistency (Wang et al., 2023) serve as our baselines, alongside a recent public multimodal process reward model, VisualPRM-8B (Wang et al., 2025).

**Training details.** To train ORMs and PRMs, we fine-tune Qwen2.5-VL-7B on our dataset as introduced in Sec. 2.4. We use the AdamW optimizer (Loshchilov & Hutter, 2019) with the weight decay as 0 and set the learning rate as $1e^{-6}$ with cosine decay. We train all models with one epoch and set the global batch size to 64. We use DeepSpeed with zero2 (Rajbhandari et al., 2020) and flash-attention-2 (Dao, 2024) to improve training efficiency. To train ORMs, a new special token `<step>` is added at the end of solutions for ORMs training. This makes a consistent formulation for ORMs and PRMs. For PRMs, `<step>` is added to separate all reasoning steps and assign labels to corresponding tokens. We use "$+$" and "$-$" as labels to denote the correctness of each step. We use $8\times$ AMD-MI250 GPUs for training and data synthesis.

**Results.** Table 1 demonstrates that Athena-ORM and Athena-PRM generally enhance the reasoning performance of MLLMs across policy models of different sizes and benchmarks. Notably, Athena-PRM achieves a **+10.2** points improvement on the WeMath dataset with Qwen2.5-VL-7B (Bai et al., 2025) as the policy model, compared to the zero-shot baseline. ORMs also exhibit general improvement across all benchmarks and policy models, although some gains are limited, such as

---

[2]All models used in this paper are "instruct" version. For simplicity, we ignore "instruct" in the text.

[3]https://huggingface.co/mistralai/Ministral-8B-Instruct-2410

Table 1: Results on seven multimodal reasoning benchmarks under Best-of-N (N=8) evaluation. †
denotes that the results are from Wang et al. (2025). We mark our results and highlight the **best**
result.

| | WeMath | MathVista | MathVision | MathVerse | DynaMath | MMMU | LogicVista |
|---|---|---|---|---|---|---|---|
| Qwen2.5-VL-7B (Bai et al., 2025) | 36.2 | 68.1 | 25.4 | 41.1 | 21.8 | 58.0 | 47.9 |
| Self-consistency (Wang et al., 2023) | 44.7 | 71.6 | 28.6 | 43.7 | 22.9 | 60.1 | 49.5 |
| VisualPRM-8B† (Wang et al., 2025) | 39.8 | 70.3 | 31.3 | 44.3 | 23.0 | 58.6 | 48.3 |
| Athena-ORM | 45.1 | 72.8 | 29.8 | 44.1 | 23.1 | 62.7 | 51.3 |
| Athena-PRM | **46.4** | **75.2** | **32.5** | **46.3** | **23.4** | **63.8** | **53.0** |
| InternVL2.5-8B (Chen et al., 2024) | 23.5 | 64.5 | 17.0 | 22.8 | 9.4 | 56.2 | 36.0 |
| Self-consistency (Wang et al., 2023) | 28.4 | 66.1 | 21.1 | 24.7 | 13.8 | 57.8 | 40.2 |
| VisualPRM-8B† (Wang et al., 2025) | **36.5** | 68.5 | **25.7** | **35.8** | 18.0 | 60.2 | 43.8 |
| Athena-ORM | 28.6 | 66.9 | 22.0 | 25.8 | 15.2 | 59.1 | 40.8 |
| Athena-PRM | 30.1 | **71.4** | 23.4 | 26.1 | **18.7** | **60.3** | **44.4** |
| Qwen2.5-VL-72B (Bai et al., 2025) | 49.1 | 74.2 | 39.3 | 47.3 | 35.9 | 70.2 | 55.7 |
| Self-consistency (Wang et al., 2023) | 54.8 | 77.0 | 43.1 | 50.8 | 37.6 | 71.1 | 59.6 |
| Athena-ORM | 55.6 | 77.8 | 43.0 | 51.2 | 39.6 | 72.3 | 60.1 |
| Athena-PRM | **58.7** | **79.1** | **44.8** | **54.6** | **42.5** | **75.8** | **60.9** |

Table 2: Results on text-only math benchmarks under Best-of-N (N=8) evaluation. † denotes that the
results are from Wang et al. (2025). We mark our results and highlight the **best** result.

(a) Qwen2.5-7B

| | GSM8K | MATH |
|---|---|---|
| Qwen2.5-7B (Yang et al., 2024) | 91.6 | 75.5 |
| Self-consistency (Wang et al., 2023) | 93.4 | 79.9 |
| VisualPRM-8B† (Wang et al., 2025) | 94.5 | 81.6 |
| Athena-ORM | 94.0 | 81.1 |
| Athena-PRM | **94.8** | **82.4** |

(b) Ministral-8B

| | GSM8K | MATH |
|---|---|---|
| Ministral-8B | 85.6 | 54.5 |
| Self-consistency (Wang et al., 2023) | 90.7 | 62.0 |
| Athena-ORM | 91.4 | 63.2 |
| Athena-PRM | **93.1** | **65.4** |

(c) Qwen2.5-72B

| | GSM8K | MATH |
|---|---|---|
| Qwen2.5-72B (Yang et al., 2024) | 95.8 | 83.1 |
| Self-consistency (Wang et al., 2023) | 96.0 | 86.0 |
| VisualPRM-8B† (Wang et al., 2025) | 96.5 | 85.2 |
| Athena-ORM | 96.0 | 86.2 |
| Athena-PRM | **97.0** | **87.4** |

+0.4 points on MathVerse with Qwen2.5-VL-72B (Bai et al., 2025). In contrast, PRMs consistently
show substantial performance gains over ORMs, illustrating the benefits of fine-grained rewards in
test-time scaling.

Table 2 highlights that Athena-PRM enhances reasoning abilities for text-only inputs in both the
Qwen2.5 series and Ministral-8B models. Specifically, Athena-PRM achieves ~~**+9.9 points**~~**+10.9
points** improvement for Ministral-8B on the challenging MATH (Hendrycks et al., 2021) benchmark,
demonstrating its effectiveness in text-only scenarios.

## 3.2 EVALUATION ON DIRECT JUDGMENT OF STEPS

**Setup.** In addition to Best-of-N evaluation, we assess Athena-PRM on VisualProcessBench (Wang
et al., 2025), comparing it with general MLLMs and VisualPRM-8B (Wang et al., 2025). VisualPro-
cessBench (Wang et al., 2025) aims to evaluate the ability to directly judge the correctness of each
step. For MLLMs, we prompt the model to analyze and judge the correctness of each step. We report
the F1-score for every subset and the micro-F1 score overall.

**Results.** The results of VisualProcessBench are presented in Table 3. It is evident that current general
open-source MLLMs struggle to judge the correctness of reasoning steps, with most ~7B models
performing no better than the random guessing baseline. Conversely, multimodal process reward
models achieve superior performance, rivaling proprietary models. Notably, our process reward
model Athena-PRM, with only 7B parameters, achieves a **+3.9** points improvement compared to
VisualPRM-8B (Wang et al., 2025), even outperforming proprietary models as judges.

## 3.3 EVALUATION ON THE FINE-TUNED MODEL

**Setup.** We select Qwen2.5-VL-7B (Bai et al., 2025) as the initial policy model, obtaining training
data as outlined in Sec. 2.5. We use AdamW (Loshchilov & Hutter, 2019) optimizer and set the
learning rate as $1e^{-6}$ with cosine decay. The batch size is set to 128, and the number of epochs is set
to one. We denote our fine-tuned model as Athena-7B and evaluate it on the same benchmarks as
Sec. 3.1 in the zero-shot setting. For evaluation, we use VLMEvalKit (Duan et al., 2024) to evaluate
Athena-7B and report the results of all baselines from Open LMM Reasoning Leaderboard.

Table 3: Results on VisualProcessBench. We report the F1-score on different dataset and macro F1 in overall. We highlight the **best** result across all models.

|  | MMMU | MathVision | MathVerse | DynaMath | WeMath | Overall |
|---|---|---|---|---|---|---|
| Random Guessing | 50.0 | 50.0 | 50.0 | 50.0 | 50.0 | 50.0 |
| *Proprietary Models as Judge* | | | | | | |
| GPT-4o-mini (OpenAI, 2024) | 53.6 | 58.9 | 57.1 | 56.7 | 58.5 | 57.9 |
| GPT-4o (OpenAI, 2024) | 56.3 | 60.2 | 59.7 | 59.0 | 63.3 | 60.3 |
| Gemini-2.0-Flash (Gemini Team, 2023) | 58.5 | 60.1 | **62.8** | 66.7 | 58.7 | 62.3 |
| *Open-source Models as Judge* | | | | | | |
| MiniCPM-V2.6-8B (Yao et al., 2025) | 44.9 | 50.9 | 58.9 | 46.7 | 57.4 | 50.4 |
| LLaVA-OV-7B (Li et al., 2025) | 45.7 | 43.0 | 42.2 | 44.7 | 52.5 | 44.4 |
| LLaVA-OV-72B (Li et al., 2025) | 46.1 | 48.4 | 53.0 | 57.0 | 57.3 | 52.3 |
| Qwen2.5-VL-7B (Bai et al., 2025) | 53.1 | 51.8 | 47.8 | 51.3 | 54.2 | 51.0 |
| Qwen2.5-VL-72B (Bai et al., 2025) | 59.2 | 59.0 | 59.7 | 62.9 | 62.3 | 60.5 |
| InternVL2.5-8B (Chen et al., 2024) | 47.1 | 45.5 | 47.8 | 50.3 | 50.8 | 48.0 |
| InternVL2.5-26B (Chen et al., 2024) | 48.8 | 47.4 | 49.2 | 50.4 | 51.4 | 49.2 |
| InternVL2.5-38B (Chen et al., 2024) | 51.5 | 48.4 | 50.9 | 51.8 | 52.5 | 50.8 |
| InternVL2.5-78B (Chen et al., 2024) | 52.0 | 51.7 | 53.7 | 50.8 | 52.5 | 52.6 |
| *Multimodal PRMs* | | | | | | |
| VisualPRM-8B (Wang et al., 2025) | 58.5 | **62.1** | 61.0 | 62.7 | 61.8 | 62.0 |
| Athena-PRM | **74.1** | 61.2 | 60.7 | **72.7** | **73.8** | **65.9** |

Table 4: Comparison of multimodal reasoning and mathematical performance across different models. Δ denotes improvements compared to Qwen2.5-VL-7B (Yang et al., 2024). We highlight improvements at least +0.5 points with **green**.

|  | MMMU | WeMath | MathVista | MathVision | MathVerse | DynaMath | LogicVista |
|---|---|---|---|---|---|---|---|
| *Proprietary Models* | | | | | | | |
| GPT-4o (OpenAI, 2024) | 72.9 | 50.6 | 71.6 | 43.8 | 49.9 | 48.5 | 64.4 |
| Claude-3.7-Sonnet (Anthropic, 2025) | 71.0 | 49.3 | 66.8 | 41.9 | 46.7 | 39.7 | 58.2 |
| Gemini-2.0-Pro (Gemini Team, 2023) | 72.6 | 56.5 | 71.3 | 48.1 | 67.3 | 43.3 | 53.2 |
| *Open-source Models (>70B)* | | | | | | | |
| InternVL2.5-78B-MPO (Wang et al., 2024c) | 68.2 | 37.6 | 76.6 | 36.2 | 43.7 | 21.2 | 50.8 |
| InternVL2.5-78B (Chen et al., 2024) | 70.1 | 39.8 | 70.6 | 32.2 | 39.2 | 19.2 | 49.0 |
| Qwen2.5-VL-72B (Bai et al., 2025) | 68.2 | 49.1 | 74.2 | 39.3 | 47.3 | 35.9 | 55.7 |
| *Open-source Models (~7B)* | | | | | | | |
| InternVL2.5-8B (Chen et al., 2024) | 56.2 | 20.2 | 58.3 | 20.0 | 20.4 | 9.2 | 33.6 |
| MiniCPM-o-2.6-8B (Yao et al., 2025) | 50.9 | 25.2 | 73.3 | 21.7 | 35.0 | 10.4 | 36.0 |
| Qwen2.5-VL-7B (Bai et al., 2025) | 58.0 | 36.2 | 68.1 | 25.4 | 41.1 | 21.8 | 47.9 |
| Athena-7B | 61.1 | 43.0 | 71.4 | 25.7 | 45.7 | 21.9 | 51.3 |
| Δ | **+3.1** | **+6.8** | **+3.3** | +0.3 | **+4.6** | +0.1 | **+3.4** |

**Results.** As shown in Table 4, Athena-7B consistently improves upon Qwen2.5-VL-7B (Bai et al., 2025) across all benchmarks, with significant enhancements of at least +0.5 points on five benchmarks. Athena-7B outperforms large open-source models (e.g. InternVL2.5-78B (Chen et al., 2024)) on five benchmarks. On the MathVista (Lu et al., 2024b), Athena-7B even achieves the comparable performance with proprietary models such as Gemini-2.0-Pro (Gemini Team, 2023), Claude-3.7-Sonnet (Anthropic, 2025), and GPT-4o (OpenAI, 2024).

## 3.4 ANALYSIS

**Test-time scaling.** To evaluate the scalability of Athena-ORM and Athena-PRM, we compare their performance using different numbers of samples ranging from 4 to 64 on the MathVista (Lu et al., 2024b). As shown in Figure 2, reward ranking for solutions at test-time improves reasoning performance across different policy models and number of samples. Specifically, ORMs and PRMs improve the self-consistency (Wang et al., 2023) baseline by 0.8 and 2.1 points with Qwen2.5-VL-72B (Bai et al., 2025) using eight samples, respectively. As the sample size increases, the performance improves steadily. However, the performance gain of ORMs is markedly lower than that of PRMs. For instance, Athena-PRM achieves 82.4% with Qwen2.5-VL-72B (Bai et al., 2025) using 64 samples, while Athena-ORM achieves similar performance with self-consistency (Wang et al., 2023) baseline. These findings underscore the advancements and scalability of PRMs in test-time scaling scenarios.

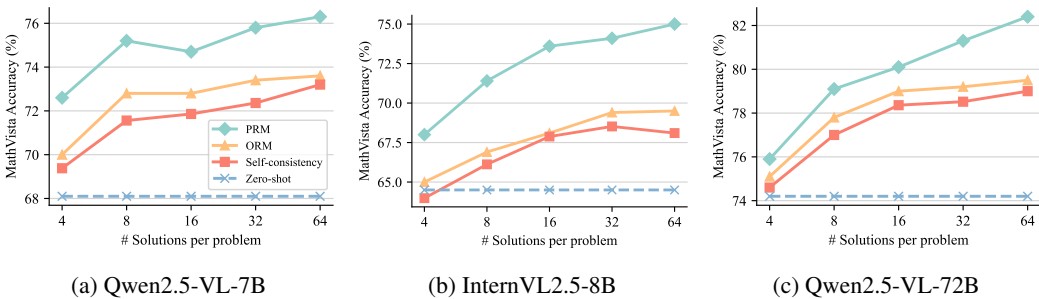

(a) Qwen2.5-VL-7B        (b) InternVL2.5-8B        (c) Qwen2.5-VL-72B

Figure 2: Best-of-N results on the MathVista (Lu et al., 2024b) across different policies. The number of solutions we sample is from 4 to 64.

Table 5: Ablation study of our design choices on the MathVista (Wang et al., 2024a) and We-Math (Qiao et al., 2025) with Qwen2.5-VL-7B (Bai et al., 2025) as the policy model.

| Method | Data | ORM init | Up-sample | MathVista | WeMath |
|---|---|---|---|---|---|
| Zero-shot | - | - | - | 68.1 | 36.2 |
| Self-consistency | - | - | - | 71.6 | 44.7 |
| Athena-ORM | Athena-600K | - | - | 72.8 | 45.1 |
| Athena-PRM | Random 5K | ✗ | ✗ | 73.1 | 45.2 |
| | Athena-5K | ✗ | ✗ | 74.1 | 45.8 |
| | Random 5K | ✔ | ✗ | 73.6 | 45.4 |
| | Athena-5K | ✔ | ✗ | 74.8 | 46.2 |
| | Athena-5K | ✔ | ✔ | **75.2** | **46.4** |
| | MC-300K | ✔ | ✔ | 75.5 | 46.4 |

**The effectiveness of our design choices.** Table 5 presents results under various design choices. Our findings indicate that PRMs trained with selectively chosen 5K samples significantly outperform those trained with randomly selected samples, achieving scores 74.1 *vs.*73.1 and 74.8 *vs.*73.6 when initialized from ORMs. Moreover, a large-scale dataset of 300K samples labeled using vanilla MC-based methods yields similar performance to our selected 5K samples. With these selected samples, ORM initialization enhances performance by 0.7 points (74.8 *vs.*74.1), and up-sampling negative labels further boosts performance by 0.4 points on the MathVista (Lu et al., 2024b).

**Computation cost analysis.** Vanilla MC-based estimation (Wang et al., 2024b) incurs substantial computational demands to estimate the correctness of intermediate steps. Our prediction consistency filter strategy significantly reduces this burden, requiring only 5K samples for training, thereby lowering the computational cost by a factor of 60. For data synthesis, we employ vLLM primarily (Kwon et al., 2023) to accelerate MLLM inference. Compared to the MC-300K dataset, our Athena-5K dataset reduces GPU hours by approximately 45-fold while achieving similar performance.

To show the scaling ability of the proposed consistency filter approach, we scale data from 5K to 60K annotated by consistency filter. We add self-consistency, ORM, PRM trained with MC-3OOK and Pass (Pass means the question is addressed if at last one solution is correct, which is the upper bound in the Best-of-N evaluation.) as baselines, and results are shown in Figure 3. From Figure 3, we could see that the performance is growing until 60K samples. Although the model trained with 5K samples is slightly worse than the model trained with MC-300K, scaling data to 10K outperforms. Results show that the proposed consistency filter approach is scalable.

# 4 CONCLUSION

In this paper, we introduce Athena-PRM, trained on 5K high-quality data with process labels. To automatically obtain accurate process labels, we employ a data filtering method based on consistency between weak and strong completers. This strategy significantly reduces computational costs for training and data synthesis while ensuring precise process labels for training PRMs. Additionally, we present two effective strategies for training PRMs: ORM initialization and negative data up-sampling.

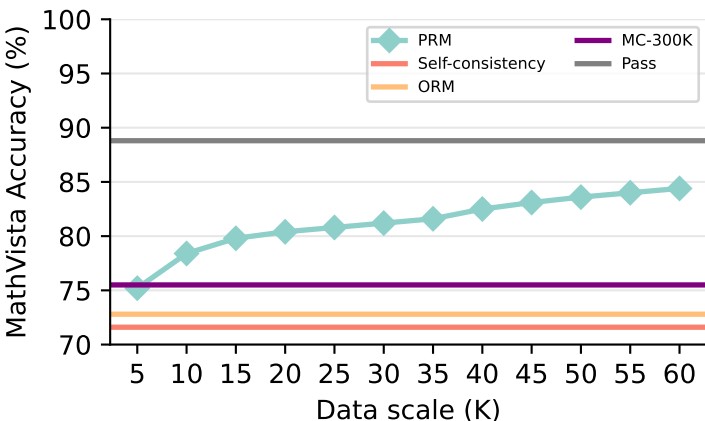

Figure 3: **Results of data scaling** from 5K to 60K. The number of sampled solutions per question is set to 8.

Athena-PRM demonstrates substantial performance improvements in Best-of-N evaluations, achieving state-of-the-art results on VisualProcessBench. Leveraging Athena-PRM, we train Athena-7B, fine-tuned from Qwen2.5-VL-7B using a reward ranked fine-tuning approach, markedly enhancing the reasoning capabilities of Qwen2.5-VL-7B. Extensive experiments confirm the effectiveness of our proposed methods.

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

## A  THE USE OF LARGE LANGUAGE MODELS (LLMS)

We only use LLMs to polish our drafts, and LLMs do not evolve research ideation.

## B  RELATED WORK

**Test-time scaling.** While scaling data and model parameters during training has been extensively studied (Kaplan et al., 2020; Hoffmann et al., 2022), scaling computation at test-time has garnered interest more recently (Snell et al., 2025; Setlur et al., 2025). TTS enables language models to leverage additional computational resources when faced with challenging questions. Our study primarily focuses on parallel scaling (Zhang et al., 2025a), wherein language models generate multiple outputs simultaneously and aggregate them into a final answer. This aggregation can involve selecting the most common outcome (Wang et al., 2023) or using reward models (Snell et al., 2025) to rank solutions, selecting the one with the highest reward, as demonstrated in our Best-of-N settings.

**Process reward models.** Reward models are crucial in reinforcement learning (Guo et al., 2025; Shao et al., 2024; Wang et al., 2024b; Ouyang et al., 2022; Bai et al., 2022; Dong et al., 2024) and test-time scaling (Setlur et al., 2025; Lightman et al., 2024; Wang et al., 2024b; Cobbe et al., 2021; Snell et al., 2025). Outcome reward models (Cobbe et al., 2021) assign a scalar value to question-response pairs, whereas process reward models evaluate each step, typically achieving superior performance and generalization (Lightman et al., 2024; Luo et al., 2024; Setlur et al., 2025; Li & Li, 2025). While better performance and generalization are achieved, collecting data to train PRMs is challenging. PRM800K (Lightman et al., 2024) is the first open-source process supervision dataset annotated by humans, but scaling it is challenging due to the time and skill required for annotating reasoning steps. Math-Shepherd (Wang et al., 2024b) proposes an automated method for estimating intermediate step correctness using Monte Carlo estimation, though it generates some incorrect labels and demands extensive computational resources. Our work explores strategies to reduce computational costs while improving process label accuracy.

## C  MORE EXPERIMENTS AND DETAILS

### C.1  CHOICES OF COMPLETERS

Here we verify the effect of combining different completers on the accuracy of estimated labels. Results are listed in Table 6. From Table 6, it is noted that the consistency filter for two completers usually improves the accuracy of estimated labels. For example, combining two weak or strong completers improves accuracy by almost two points in general. Furthermore, the accuracy of estimated labels is improved by combining one weak and one strong completer, achieving the most competitive accuracy, e.g., 95.2% with weak completer $\phi_w^2$ and strong completer $\phi_s^1$. This shows that the proposed consistency filter strategy is effective across all combinations of different completers and achieves the best performance when we combine a weak completer and a strong completer. We also explore use more completers to further improve the qualify of process labels. However, we did not observe any significant improvement, despite the increased computational cost.

### C.2  BENCHMARKS

We provide more details about used benchmarks in Table 7.

### C.3  REWARD CHOICE

In this paper, we consider three reward choices of Athena-PRM for aggregating step scores into a final score (Zhang et al., 2025b), including PRM-minium, PRM-last, and PRM-product. For all reasoning steps $\{a_i\}_{i=1}^K$, its corresponding reward for each step is $\{r_i\}_{i=1}^K$, the reward $r$ for each solution with different scoring methods are calculated as follows: 1) PRM-minium uses the minium reward across all steps, i.e., $r = \min\{r_i\}_{i=1}^K$; 2) PRM-last uses the reward of last step, i.e., $r = r_K$; 3) PRM-product uses the reward product of all steps, i.e. $r = \prod_{i=1}^K r_i$.

Table 6: The accuracy of estimated process labels with Different completers combination. In Two Completers, we use consistency filter introduced in Sec. 2.2. $\phi_w^1$, $\phi_w^2$, $\phi_s^1$, and $\phi_w^2 \phi_s^2$ denote Mistral-7B-Instruct, Qwen2-7B-Instruct, Qwen2.5-72B-Instruct and Llama-3-70B-Instruct, respectively.

| | Weak Completer | | Strong Completer | | Accuracy(%) |
|---|---|---|---|---|---|
| | $\phi_w^1$ | $\phi_w^2$ | $\phi_s^1$ | $\phi_s^2$ | |
| Single Completer | ✔ | | | | 78.2 |
| | | ✔ | | | 80.1 |
| | | | ✔ | | 83.4 |
| | | | | ✔ | 84.7 |
| Two Completers | ✔ | ✔ | | | 82.3 |
| | | | ✔ | ✔ | 85.6 |
| | ✔ | | ✔ | | 94.1 |
| | ✔ | | | ✔ | 93.8 |
| | | ✔ | ✔ | | 95.2 |
| | | ✔ | | ✔ | 94.7 |

Table 7: Details of all benchmarks used and corresponding metrics.

| | Split | # Samples | Metric |
|---|---|---|---|
| MathVista (Lu et al., 2024b) | Testmini | 1000 | Accuracy |
| MathVision (Wang et al., 2024a) | Full | 3040 | Accuracy |
| MathVerse (Zhang et al., 2024) | Vision-Only | 788 | Accuracy |
| DynaMath (Zou et al., 2025) | Full | 5050 | Accuracy |
| WeMath (Qiao et al., 2025) | Testmini | 1740 | Score (Strict) |
| LogicVista (Xiao et al., 2024) | Full | 448 | Accuracy |
| MMMU (Yue et al., 2024) | Validation | 900 | Accuracy |
| GSM8K (Cobbe et al., 2021) | Test | 1319 | Accuracy |
| MATH (Hendrycks et al., 2021) | Test | 5000 | Accuracy |
| VisualProcessBench (Wang et al., 2025) | - | 2866 | F1-score |

We list results in Table 8. It is shown that different choices of selecting a reward for PRMs could produce different performance. In all other experiments, we use the minimum reward across all steps as the reward for the solution unless otherwise specified.

Table 8: Best-of-8 performance comparison on the MathVista (Lu et al., 2024b) with Qwen2.5-VL-7B (Bai et al., 2025) with different reward choices: last step reward, minium reward across all steps and product of all rewards. Default settings are marked in gray.

| Method | Accuracy(%) |
|---|---|
| Zero-shot | 68.1 |
| Self-consistency | 71.6 |
| ORM | 72.8 |
| PRM-product | 74.5 |
| PRM-last | 75.1 |
| PRM-minium | **75.2** |

## C.4 LABEL DISTRIBUTION

We present the distribution of process labels in Table 9. It is shown that *label imbalance* exists in PRM800K and Athena-300K. We try different up-sample rates (e.g. ×2) for samples with error steps and find that slightly up-sampling negative data is beneficial for training PRMs, as shown in Table 10. We find that increasing the up-sample rate will not improve performance. We set the up-sample rate to 2 for other experiments.

Table 9: Distribution of labels across different datasets. Good, neutral, and bad denote different process labels.

| Dataset | Good | Neutral | Bad |
|---|---|---|---|
| PRM800K (Lightman et al., 2024) | 76% | 12% | 12% |
| AthenaMC-300K | 81% | - | 19% |
| Athena-5K | 79% | - | 21% |

Table 10: Results on MathVista under Best-of-8 setting with different up-sampling rate. We use Athena-5K as training dataset and Qwen-2.5-VL-7B as the policy model. Default settings are marked in gray.

| Method | Up-sample rate | Accuracy(%) |
|---|---|---|
| Zero-shot | - | 68.1 |
| Self-consistency | - | 71.6 |
| Athena-ORM | - | 72.8 |
| Athena-PRM | ✗ | 74.8 |
| Athena-PRM | ×2 | **75.2** |
| Athena-PRM | ×5 | **75.2** |
| Athena-PRM | ×10 | 74.9 |

## C.5 EFFECTIVENESS OF ATHENA-PRM FOR REWARD RANKED FINE-TUNING

Besides selecting responses with Athena-PRM, we have tried to directly use all responses to fine-tune. We do not use Athena-ORM to select responses because all data we use in reward-ranked fine-tuning has ground-truth answers. Before we use all data to fine-tune, we conduct a quick test to verify the effectiveness of response ranking with Athena-PRM. We used a small-scale dataset ( 30k queries) to verify the effectiveness of response ranking with Athena-PRM. Results are shown in Table 11.

Table 11: Results of reward ranked fine-tuning (RAFT) with different data settings. We report accuracy on the MathVista with Qwen2.5-VL-7B.

| Method | Data | Athena-PRM | Accuracy |
|---|---|---|---|
| Qwen2.5-VL-7B | - | - | 68.1 |
| +RAFT | 30k | ✗ | 68.7 |
| +RAFT | 30k | ✔ | 69.8 |
| +RAFT (ours) | 150k | ✔ | 71.4 |

## D CASE STUDY

We provide some examples from VisualProcessBench and MathVista to help readers understand how PRMs work, see Figure 4, 5, 6, 7.

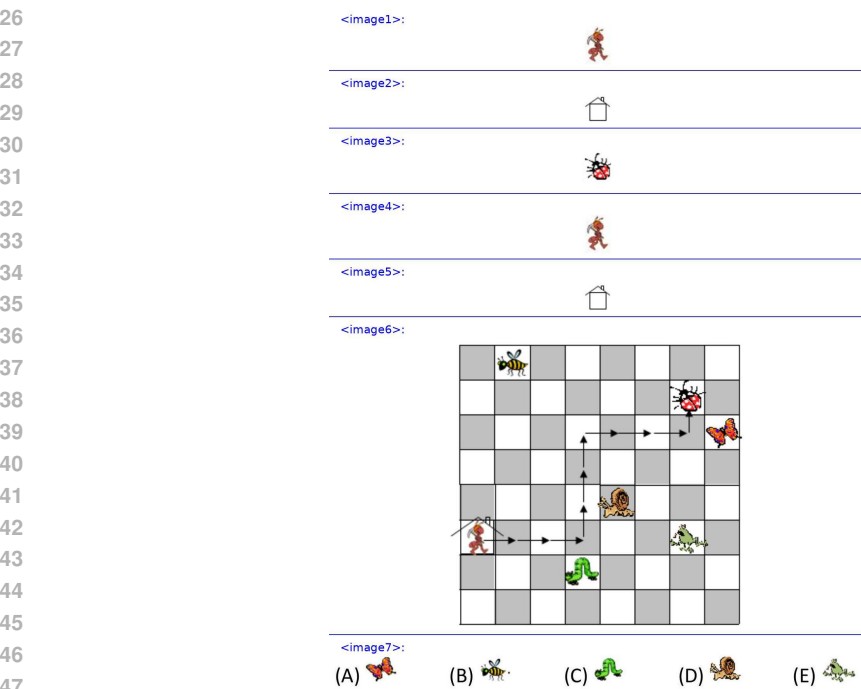

Hint: Please answer the question and provide the correct option letter, e.g., A, B, C, D, at the end.
Question: When the ant <image1> walks from home <image2> along the arrows $\rightarrow 3, \uparrow 3, \rightarrow 3, \uparrow 1$, he gets to the ladybird <image3>.
Which animal does the ant <image1> get to when he walks from home <image2> along the following arrows: $\rightarrow 2, \downarrow 2, \rightarrow 3, \uparrow 3, \rightarrow 2, \uparrow 2$?
<image6>
<image7>
Choices:(A) A (B) B (C) C (D) D (E) E
(Answer: A)

| Step | Reward | Label |
|---|---|---|
| To determine which animal the ant reaches, we will follow the given movements step-by-step on the grid: | 0.95 | ✔ |
| 1. Start at the home position ('<image2>'), which is at the bottom-left corner of the grid. | 0.99 | ✔ |
| 2. Follow the arrows:
- $\rightarrow 2$: Move 2 squares to the right.
- $\downarrow 2$: Move 2 squares down.
- $\rightarrow 3$: Move 3 squares to the right.
- $\uparrow 3$: Move 3 squares up.
- $\rightarrow 2$: Move 2 squares to the right.
- $\uparrow 2$: Move 2 squares up. | 0.99 | ✔ |
| 3. Trace the movement:
- Start at the bottom-left corner.
- Move 2 squares right: You are now in the third column of the first row.
- Move 2 squares down: You are now in the third column of the third row.
- Move 3 squares right: You are now in the sixth column of the third row.
- Move 3 squares up: You are now in the sixth column of the top row.
- Move 2 squares right: You are now in the eighth column of the top row.
- Move 2 squares up: You are now in the eighth column of the top row (still at the butterfly position). | 0.12 | ✗ |
| The ant reaches the **butterfly**, which corresponds to **option A**. | 0.87 | ✔ |
| ### Final Answer: **A** | 0.75 | ✔ |

Figure 4: A case study from VisualProcessBench (Wang et al., 2025).

As shown in the figure, passing point C to draw the tangent of circle O. then the degree of angle A is ()
Choices: A:20°   B:25°   C:.40°   D:50°

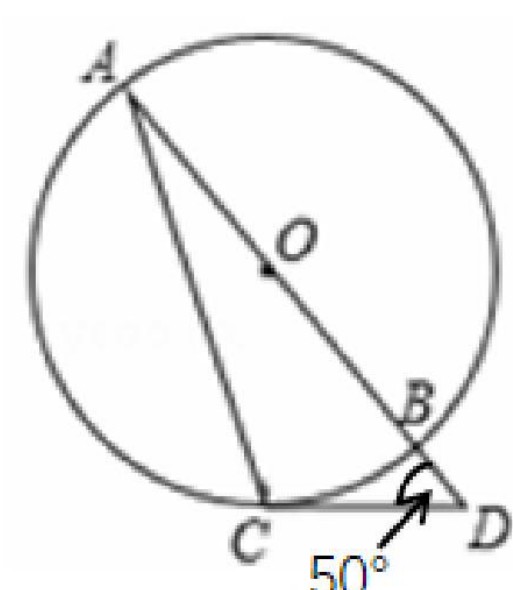

According to the question shown in the image, please directly answer the question and provide the correct option letter, e.g., A, B, C, D.
(Answer: A)

| Step | Reward | Label |
|---|---|---|
| To find the degree of angle A, we can use the property that the angle subtended by an arc at the center of the circle is twice the angle subtended at any point on the circle. | 0.82 | ✔ |
| 1. **Identify the key points:** 
 - The arcs are AC and BC. 
 - Angle A is subtended by these arcs at point O (the center of the circle). | 0.85 | ✔ |
| 2. **Understand the given information:** 
 - CD is a tangent at point C. 
 - The angle between the tangent and the chord is equal to the angle subtended by the chord at the circumference. Therefore, angle CDB = 50°. | 0.85 | ✔ |
| 3. **Angle properties:** 
 - The angle A is subtended by arc BC at the center. 
 - Since CD is a tangent, angle A = 50° (as the inscribed angle is half the central angle). | 0.23 | ✘ |
| Thus, the degree of angle A is 20°. | 0.67 | ✔ |
| Final answer: A | 0.96 | ✔ |

Figure 5: A case study from VisualProcessBench (Wang et al., 2025).

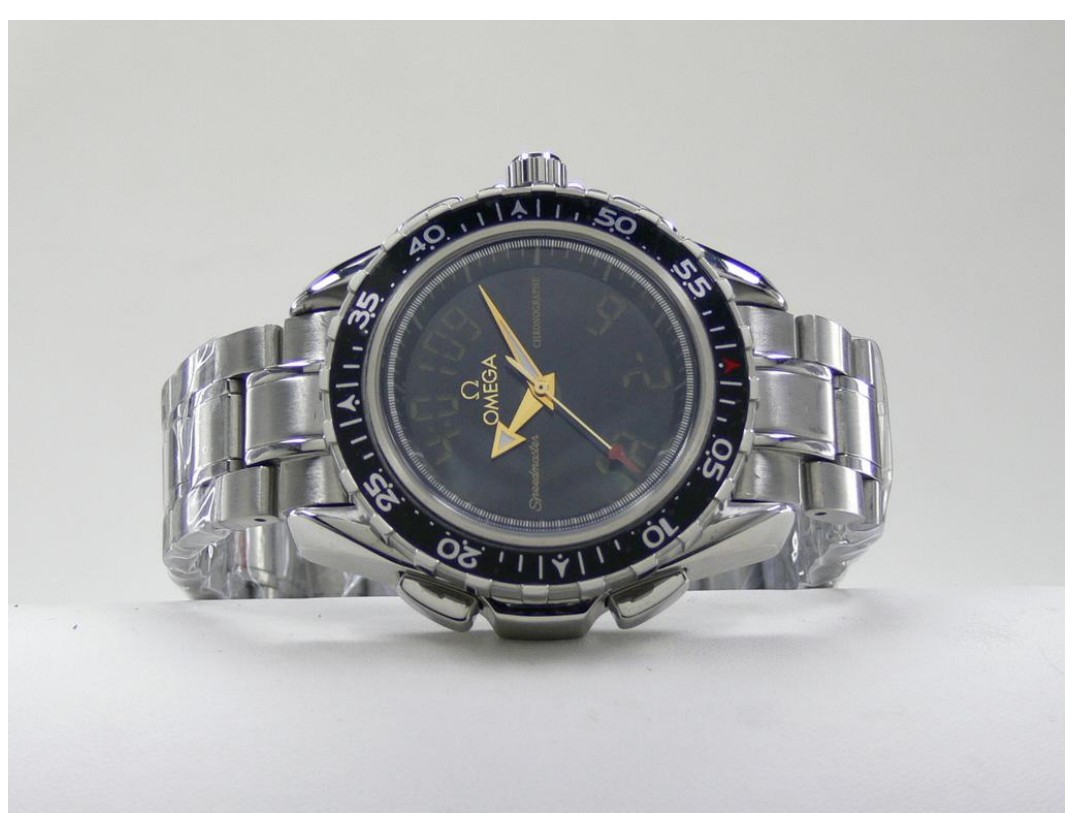

Hint: Please answer the question requiring an integer answer and provide the final value, e.g., 1, 2, 3, at the end.

Question: what is the highest number shown on the black outer part of the watch?

(Answer: 55)

| Step | Reward | Label |
|---|---|---|
| To determine the highest number shown on the black outer part of the watch, let's analyze the image step by step | 0.90 | ✔ |
| 1. The black outer part of the watch appears to be a bezel with numbers printed on it. | 0.87 | ✔ |
| 2. The numbers on the bezel are arranged in a circular pattern. | 0.96 | ✔ |
| 3. Observing the numbers, they range from 0 to 60, increasing in increments of 5. | 0.83 | ✔ |
| 4. The highest number visible on the bezel is 60. | 0.34 | ✘ |
| Therefore, the highest number shown on the black outer part of the watch is 60. | 0.15 | ✘ |
| Answer: 60 | 0.04 | ✘ |

Figure 6: A case study from MathVista (Lu et al., 2024b) with Qwen2.5-VL-72B (Bai et al., 2025).

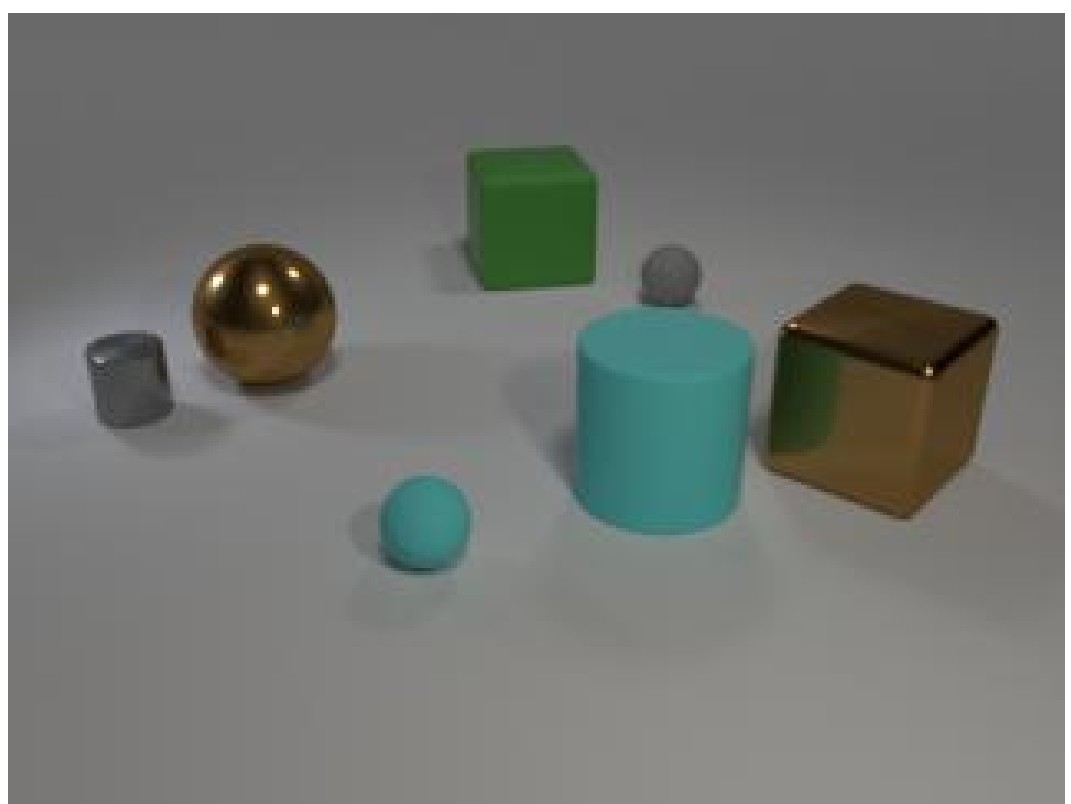

Hint: Please answer the question requiring an integer answer and provide the final value, e.g., 1, 2, 3, at the end.

Question: Subtract all large green matte cubes. Subtract all big green blocks. How many objects are left?

(Answer: 6)

| Step | Reward | Label |
|---|---|---|
| To solve this problem, let's identify and count the objects in the image: | 0.95 | ✔ |
| 1. There is one large green matte cube. | 0.99 | ✔ |
| 2. There are no other big green blocks since the only green block is already counted. | 0.90 | ✔ |
| Now, we subtract these objects as per the instructions | 0.92 | ✔ |
| - Subtract all large green matte cubes: This removes the large green matte cube from the count. | 0.99 | ✔ |
| - Subtract all big green blocks: Since there are no other big green blocks, nothing additional is removed. | 0.99 | ✔ |
| After removing the large green matte cube, the remaining objects are:
- A small gray metallic cylinder.
- A large gold metallic sphere.
- A small gray matte sphere.
- A large cyan matte cylinder.
- A small cyan matte sphere.
- A large gold metallic cube. | 0.84 | ✔ |
| Counting the remaining objects, we have 6 objects left. | 0.92 | ✔ |
| Answer: 6 | 0.99 | ✔ |

Figure 7: A case study from MathVista (Lu et al., 2024b) with Qwen2.5-VL-72B (Bai et al., 2025).

