# OpenReview forum: "Athena: Enhancing Multimodal Reasoning with Data-efficient Process Reward Models"
_ICLR.cc/2026/Conference — Submitted to ICLR 2026_

### Official Review · Reviewer_4HPA · 2025-10-29

**Soundness:** 2
**Presentation:** 3
**Contribution:** 3
**Rating:** 4
**Confidence:** 4

**Summary:**

This paper introduces Athena-PRM, a multimodal process reward model (PRM) designed to evaluate the reward scores of reasoning steps within complex tasks. The authors assert that annotation costs can be reduced by generating pseudo step-level labels through the predictive consistency between weak and strong completers. To further enhance PRM performance, the study proposes ORM initialization and negative data upsampling. Experimental results across multiple reasoning benchmarks reportedly surpass existing models, with the proposed method achieving state-of-the-art outcomes on both the VisualProcessBench and math-related datasets.

**Strengths:**

This paper addresses the cost and scalability issues of step-level reward labeling by weak-strong completers, a key challenge in process supervision. This work also demonstrates significant improvements across several benchmark datasets.

**Weaknesses:**

- The authors repeatedly emphasize the results obtained using 5,000 samples but fail to specify the complexity of each sample or the cost of generating completions from the strong and weak models. This makes the argument for efficiency unconvincing.
- The data appears to be cherry-picked from the 60k sample pool, as there is minimal performance gain when scaling from 5k to 60k samples. An analysis of the performance growth curve is needed to validate the scaling properties.
- There is a discrepancy between the model used for validation within the strong-weak model framework and the actual target model used in the experiments. Therefore, it remains questionable whether the method is genuinely effective for the target model.
- The paper lacks a comparative analysis with other data synthesis strategies and does not include ablation studies on relevant components. Consequently, it is unclear how much the core data sampling strategy actually contributes to the observed performance.
- The experiments are primarily focused on mathematical reasoning datasets (e.g., WeMath, MathVista). It is worth investigating whether the method can still generate high-quality Best-of-N (BoN) samples for reasoning tasks in other domains, such as commonsense and science (e.g., by testing on benchmarks like M3CoT or EMMA).
- The study appears to exclusively use Qwen2.5-VL-7B as the training backbone, which makes it difficult to assess the general applicability of the data. This approach risks biasing the evaluation towards the capabilities of a specific large model rather than the intrinsic quality of the PRM data itself.

> My inquiry involves multiple questions regarding an extensive set of experiments. I will adjust the score to reflect the degree to which your response resolves these points.

**Questions:**

- Could the authors quantify the complexity of each sample and provide the computational cost associated with generating completions from both the strong and weak models? Without such data, the argument for the method's efficiency is difficult to substantiate.
- Could the authors present a performance growth curve for scaling from 5,000 to 60,000 samples? The minimal performance gain observed suggests a potential selection bias in the data, and an analysis of scaling properties would be necessary to validate the method's scalability.
- A discrepancy is noted between the model used for validation within the strong-weak framework and the target model employed in the final experiments. Could the authors justify this choice and provide evidence that the method's effectiveness indeed transfers to the final target model?
- To what extent does the core data sampling strategy contribute to the observed performance gains? A comparative analysis against other data synthesis methods, would be crucial to isolate the contribution of the proposed technique.
- How effective is it to use data distilled solely from a weak or a strong completer?
- Given the experimental focus on mathematical reasoning, it is unclear whether the method can generate high-quality Best-of-N (BoN) samples for reasoning tasks in other domains. Have the authors considered evaluating its efficacy on commonsense or scientific reasoning benchmarks, such as M3CoT or EMMA?
- The study's reliance on Qwen2.5-VL-7B as the sole training backbone makes it difficult to assess the general applicability of the data. Have the authors considered experiments with other model architectures to disentangle the intrinsic quality of the PRM data from the capabilities of this specific model?

---

> ### Author Response · Authors · 2025-11-19
> **Response to 4HPA**
>
> We’d like to thank you for the comprehensive and constructive comments, and we now address each of the given concerns:
>
> > (W5 & Q6) Experiments on Reasoning Tasks with Other Dataset.
>
> We conducted more experiments on M3CoT (dev and test split) [1] and EMMA (on the whole dataset) [2] under the Best-of-N (N=8) setting. Results are summarized as follows.
>
> |Method | M3CoT-dev | M3CoT-test |EMMA|
> |:--| :---: | :---:| :---: |
> | Qwen2.5-VL-7B | 62.9 | 61.7 | 23.4 |
> | Self-consistency | 67.5 | 66.5 | 25.8 |
> | Athena-ORM | 68.4 | 69.0  | 27.4 |
> | Athena-PRM | 72.2| 72.8 | 29.1 |
>
> Results show that Athena-PRM still significantly enhances the reasoning performance on other domains.
>
> > Q5:  How effective is it to use data distilled solely from a weak or a strong completer?
>
> We conduct experiments using data whose response is distilled from the strong completer (i.e., Qwen2.5-VL-72B).
>
> | Model | MMMU | WeMath | MathVista | MathVision | MathVerse | DynaMath | LogicVista |
> |---|---|---|---|---|---|---|---|
> | Qwen2.5-VL-7B | 58.0 | 36.2 | 68.1 | 25.4 | 41.1 | 21.8 | 47.9 |
> | Athena-7B | 61.1 | 43.0 | 71.4 | 25.7 | 45.7 | 21.9 | 51.3 |
> | Athena-7B-distill | 58.8 | 41.7 | 70.6 | 25.4 |44.6 |21.9 |50.8|
>
> Results show that for the same dataset, distillation is worse than reward-ranked fine-tuning with PRM. We analyze that filter response with distillation lacks intermediate verification compared with reward-ranked fine-tuning with PRMs.
>
> > W1 & Q1: The authors repeatedly emphasize the results obtained using 5,000 samples but fail to specify the complexity of each sample or the cost of generating completions from the strong and weak models. This makes the argument for efficiency unconvincing.
>
> We provide computation cost analysis in Sec.3.4. We use GPU hours as a computation cost metric.  For training PRMs, the training cost of Athena-5K is 60x less than with MC-300K. For data, generation Athena-5K uses 45x less than MC-300K. For synthesizing MC-300K, the data annotation takes approximately 20 days on 8 MI-250 GPUs, while 10 hours on 8 MI-250 GPUs for Athena-5K.
>
> > W2 & Q2: Scaling more data from 5k to 60 K
>
> We scale data from 5K to 60K. We report the performance on MathVista including 1000 questions. The number of sampled solutions is set to 8.
>
> | Method  | Accuracy |
> |:---|:---:|
> | Qwen2.5-VL-7B | 68.1 |
> | Self-consistency |71.6 |
> | Athena-ORM |72.8 |
> | PRM trained with MC-300K | 75.5 |
> |Athena-PRM-5K |75.2 |
> |Athena-PRM-10K | 78.4 |
> | Athena-PRM-15k |79.8 |
> | Athena-PRM-20k | 80.4 |
> |Athena-PRM-25k | 80.8 |
> | Athena-PRM-30k | 81.2 |
> | Athena-PRM-35k | 81.6 |
> | Athena-PRM-40k | 82.5 |
> | Athena-PRM-45k | 83.1 |
> |Athena-PRM- 50k | 83.6 |
> |Athena-PRM- 55k | 84.0 |
> |Athena-PRM-60k | 84.4 |
> |Pass (upper bound) |88.8 |
>
>
> Results show that the advantage of high-quality data will matter at scale. Scaling data from 5K to 60K, the accuracy of MathVista increases 9.2 points (75.2 to 84.4). For better clarity, we plot the curve of data and accuracy in Sec. 3.4, see our revised version.
>
>
> > W6 & Q7: other backbones for PRMs training.
>
> We choose InternVL2.5-8B as the base model to train PRMs with Athena-5K. We present results on VisualProcessBench as follows.
> | | Backbone | VisualProcessBench |
> |:----|:----:|:----:|
> Qwen2.5-VL-7B | - | 51.0 |
> InternVL2.5-8B | - | 48.0 |
> Athena-PRM(-7B) | Qwen2.5-VL-7B | 65.9 |
> VisualPRM-8B | InternVL2.5-8B | 62.0 |
> Athena-PRM(-8B) | InternVL2.5-8B |64.7 |
>
> Results show that Athena-5K trained on InternVL2.5-8B still outperforms VisualPRM-8B.
>
> > W3 & Q3: There is a discrepancy between the model used for validation within the strong-weak model framework and the actual target model used in the experiments. Therefore, it remains questionable whether the method is genuinely effective for the target model.
>
> We conduct experiments on VisualProcessBench  with Qwen2.5-VL-3B as the weak completer and Qwen2.5-VL-72B as the strong completer (the same model used in our experiments). Results are as follows:
>
> | Completer | Accuracy |
> |---|---|
> |Qwen2.5-VL-3B | 70.6 |
> | Qwen2.5-VL-72B | 74.2 |
> | Qwen2.5-VL-3B + 72B |89.9 |
>
> Results show that the proposed consistency filter strategy is still effective under different models and datasets.
>
> > W4 & Q4: Comparison with other methods and ablation.The paper lacks a comparative analysis with other data synthesis strategies and does not include ablation studies on relevant components. Consequently, it is unclear how much the core data sampling strategy actually contributes to the observed performance.
>
> For each contribution, we present ablation study in Table 5. As for comparison with other methods, we mainly compare with vanilla MC-estimation [3] in Table 5 (line MC-300K).
>
>
> Reference
> [1] M^3CoT: A Novel Benchmark for Multi-Domain Multi-step Multi-modal Chain-of-Thought. ACL 2024.
> [2] Can MLLMs Reason in Multimodality? EMMA: An Enhanced MultiModal ReAsoning Benchmark. ICML 2025.
> [3] Math-Shepherd: Verify and Reinforce LLMs Step-by-step without Human Annotations. ACL 2024.

---

> ### Author Response · Authors · 2025-11-26
>
> Dear Reviewer 4HPA,
>
> We hope this message finds you well. As the discussion period is approaching its end (about **one week** remaining), we wanted to kindly check whether you have any additional comments or concerns that we could help clarify. We truly appreciate your time and effort in reviewing our work, and we are happy to provide any further details that might assist your evaluation.
>
> Thank you again for your valuable feedback.
>
> Best regards, Authors

---

> ### Author Response · Authors · 2025-11-27
>
> Dear Reviewer 4HPA,
>
> We hope this message finds you well. As the discussion period is approaching its end (with about one week remaining), we would like to kindly ask whether our responses have adequately addressed your concerns regarding our submission.
>
> We sincerely appreciate your valuable comments and the time you have taken to review our work. We would be very happy to provide further clarifications or engage in additional discussion if you have any remaining questions or suggestions.
>
> We believe that detailed and constructive discussion is highly beneficial to the research community, and we are grateful for your contributions to this process.
>
> Thank you again for your time and consideration.
>
> Best regards
>
> Authors

---

### Official Review · Reviewer_gqQb · 2025-10-30

**Soundness:** 3
**Presentation:** 3
**Contribution:** 3
**Rating:** 8
**Confidence:** 3

**Summary:**

This paper proposes Athena-PRM, a multimodal process reward model (PRM) designed to be data-efficient. The core problem it addresses is the prohibitive cost of acquiring step-level annotations for training PRMs. The authors introduce a data-filtering method that leverages prediction consistency between a weak completer and a strong completer to identify high-quality, reliable process labels from automated methods. Using this method, they demonstrate that a PRM trained on only 5,000 high-quality samples can achieve performance comparable to or better than models trained on 300K noisy samples. The paper also introduces two effective training strategies: ORM initialization which uses a pretrained outcome reward model as a starting point and negative data up-sampling to handle label imbalance. The resulting Athena-PRM is validated across three scenarios—test-time scaling (Best-of-N), direct step evaluation, and reward-ranked fine-tuning, where it significantly boosts the performance of various policy models and achieves state-of-the-art results on the VisualProcessBench benchmark.

**Strengths:**

The "consistency filtering" technique using weak and strong completers is an intuitive and clever way to distill high-quality labels from noisy, automated sources. The finding that a mere 5K high-quality samples can rival a 300K-sample dataset is a significant contribution, addressing the bottleneck of annotation cost and computational expense in reward modeling.

The authors conduct thorough empirical validation across different application scenarios, with Athena-PRM achieving state-of-the-art performance on VisualProcessBench and even surpassing proprietary models like GPT-4o, underscoring its versatility and effectiveness.

The paper is well-structured, and its claims are well-supported by ablation studies. Table 5 clearly disentangles the contributions of the key components: the high-quality Athena-5K data, the ORM initialization, and the negative up-sampling. This analysis confirms that each proposed strategy provides a tangible benefit, making the methodology easy to follow and justifying the design choices.

**Weaknesses:**

While the PRM is evaluated on policy models from different families (Qwen, InternVL, Ministral) , the PRM itself is a fine-tuned Qwen2.5-VL-7B. Furthermore, the data generation process is dominated by the Qwen family. This raises a minor concern about generalizability. It would be more convincing if the authors showed that the Athena-5K dataset could be used to train another effective PRM based on a different model family.

**Questions:**

The ablation in Table 5 is a key result, showing Athena-5K outperforms Random-5K. How does the performance of your consistency-filtered dataset (Athena-K) scale as K increases? For instance, does an "Athena-50K" dataset significantly outperform the MC-300K baseline, or do the returns on data quality diminish quickly after the 5K-sample mark?

---

> ### Author Response · Authors · 2025-11-19
> **Response to gqQb**
>
> Thank you for the comprehensive and constructive comments and for a positive assessment of our paper. Now we address each of the given concerns:
>
> > W: While the PRM is evaluated on policy models from different families (Qwen, InternVL, Ministral) , the PRM itself is a fine-tuned Qwen2.5-VL-7B. Furthermore, the data generation process is dominated by the Qwen family. This raises a minor concern about generalizability. It would be more convincing if the authors showed that the Athena-5K dataset could be used to train another effective PRM based on a different model family.
>
> We choose InternVL2.5-8B as the base model to train PRMs with Athena-5K. We present results on VisualProcessBench as follows.
> | | Backbone | VisualProcessBench |
> |:----|:----:|:----:|
> Qwen2.5-VL-7B | - | 51.0 |
> InternVL2.5-8B | - | 48.0 |
> Athena-PRM(-7B) | Qwen2.5-VL-7B | 65.9 |
> VisualPRM-8B | InternVL2.5-8B | 62.0 |
> Athena-PRM(-8B) | InternVL2.5-8B |64.7 |
>
>
> Results show that Athena-5K trained on InterVL2.5-8B still outperforms VisualPRM-8B.
>
> >Q：The ablation in Table 5 is a key result, showing Athena-5K outperforms Random-5K. How does the performance of your consistency-filtered dataset (Athena-K) scale as K increases? For instance, does an "Athena-50K" dataset significantly outperform the MC-300K baseline, or do the returns on data quality diminish quickly after the 5K-samplemark?
>
> We scale data from 5K to 60K. We report the performance on MathVista including 1000 questions. The number of sampled solutions is set to 8.
>
> | Method  | Accuracy |
> |:---|:---:|
> | Qwen2.5-VL-7B | 68.1 |
> | Self-consistency |71.6 |
> | Athena-ORM |72.8 |
> | PRM trained with MC-300K | 75.5 |
> |Athena-PRM-5K |75.2 |
> |Athena-PRM-10K | 78.4 |
> | Athena-PRM-15k |79.8 |
> | Athena-PRM-20k | 80.4 |
> |Athena-PRM-25k | 80.8 |
> | Athena-PRM-30k | 81.2 |
> | Athena-PRM-35k | 81.6 |
> | Athena-PRM-40k | 82.5 |
> | Athena-PRM-45k | 83.1 |
> |Athena-PRM- 50k | 83.6 |
> |Athena-PRM- 55k | 84.0 |
> |Athena-PRM-60k | 84.4 |
> |Pass (upper bound) |88.8 |
>
>
> Results show that the advantage of high-quality data will matter at scale. Scaling data from 5K to 60K, the accuracy of MathVista increases 9.2 points (75.2 to 84.4). For better clarity, we plot the curve of data and accuracy in Sec. 3.4, see our revised version.

---

> > ### Comment · Reviewer_gqQb · 2025-11-21
> >
> > Thank you for the detailed response. The additional experiments and the newly included Figure 3 address my question and concern. I find the new results informative.
> >
> > I appreciate the authors’ efforts in clarifying the question and providing further evidence.

---

> > > ### Author Response · Authors · 2025-11-26
> > >
> > > We are pleased that our response has satisfactorily addressed your concerns. We appreciate your time and effort in reviewing our submission.

---

### Official Review · Reviewer_KwTQ · 2025-10-30

**Soundness:** 2
**Presentation:** 2
**Contribution:** 2
**Rating:** 4
**Confidence:** 4

**Summary:**

This paper proposes Athena-PRM, a data-efficient multimodal Process Reward Model (PRM) that scores each step in a reasoning chain. The model is leveraged in three scenarios: (1) test-time Best-of-N expansion, (2) direct step-correctness evaluation (VisualProcessBench), and (3) Reward-Ranked Fine-Tuning (RAFT) to produce Athena-7B.

**Strengths:**

The approach is well-motivated and demonstrates performance improvements.

**Weaknesses:**

While the proposed filtering strategy is the core contribution, it raises two significant concerns: Efficiency: The strategy is alarmingly inefficient, reducing 300K samples to a mere 5K. Bias: This aggressive filtering may introduce selection bias, potentially retaining only the most high-confidence correct and incorrect examples.

**Questions:**

The comparison in Table 3 is confounded. Given that the Qwen base model itself already provides a 3.0-point advantage over InternVL, the reported 3.9-point improvement for Athena-PRM cannot be cleanly attributed to the proposed method.

The baseline in Table 4 is inadequate. To properly isolate the PRM's contribution, the baseline must also be trained on the same filtered dataset, but with a randomly selected trajectory for SFT, rather than the PRM-selected one. Without this direct ablation, it is impossible to prove that the performance gain stems from the PRM's selection capability and not merely from the introduction of the filtered data itself.

Part of the paper are confusing. For example, Athena-300K  in line893 should be MC-300K?   Athena-5K should be a subset of MC-300K?+9.9 in line354 should be 10.9?

---

> ### Author Response · Authors · 2025-11-19
> **Response to KwTQ**
>
> We’d like to thank you for the comprehensive and constructive comments, and we now address each of the given concerns:
>
> >W：While the proposed filtering strategy is the core contribution, it raises two significant concerns: Efficiency: The strategy is alarmingly inefficient, reducing 300K samples to a mere 5K. Bias: This aggressive filtering may introduce selection bias, potentially retaining only the most high-confidence correct and incorrect examples.
>
> Athena-5k is not filtered from the whole 300K, it is just a subset. In fact, we only use 60K samples  from MC-300K and we obtain 5K samples using the proposed consistency filter.
>
> As for bias, we only consider the correctness of process labels, i.e. only 0/1 labels. So only using the most high-confidence correct and incorrect examples will not affect too much. We believe that few but high-quality data is more useful than much but low-quality data. We run ablation with different data in Table 5.
>
> > Q1: The comparison in Table 3 is confounded. Given that the Qwen base model itself already provides a 3.0-point advantage over InternVL, the reported 3.9-point improvement for Athena-PRM cannot be cleanly attributed to the proposed method.
>
> We fine-tune the InternVL2.5-8B (the same base model with VisualPRM-8b) using the same data with Athena-PRM (7B).  Results on VisualProcessBench are as follows:
> | | Backbone | VisualProcessBench |
> |:----|:----:|:----:|
> Qwen2.5-VL-7B | - | 51.0 |
> InternVL2.5-8B | - | 48.0 |
> Athena-PRM(-7B) | Qwen2.5-VL-7B | 65.9 |
> VisualPRM-8B | InternVL2.5-8B | 62.0 |
> Athena-PRM(-8B) | InternVL2.5-8B |64.7 |
>
> Results show that with the same backbone, Athena-PRM still outperforms VisualPRM-8B.
>
>
> >Q2: The baseline in Table 4 is inadequate. To properly isolate the PRM's contribution, the baseline must also be trained on the same filtered dataset, but with a randomly selected trajectory for SFT, rather than the PRM-selected one. Without this direct ablation, it is impossible to prove that the performance gain stems from the PRM's selection capability and not merely from the introduction of the filtered data itself.
>
> We present ablation study in Table 11 in the submission. Besides selecting responses with Athena-PRM, we have tried to directly use all responses to fine-tune. We do not use Athena-ORM to select responses because all data we use in reward-ranked fine-tuning has ground-truth answers. Before we use all data to fine-tune, we conduct a quick test to verify the effectiveness of response ranking with Athena-PRM. We used a small-scale dataset ( 30k queries) to verify the effectiveness of response ranking with Athena-PRM.  For convenience, we put the results here. We report best-of-N (N=8) results on MathVista.
>
> | Method | Data | Athena-PRM | Accuracy |
> |:--| :---:| :---: | :--: |
> | Qwen2.5-VL-7B | - | -| 68.1 |
> +RAFT| 30K| × | 68.7 |
> +RAFT | 30K |√ | 69.8 |
> +RAFT| 150K | √ |71.4 |
>
> It is shown that select responses with Athena-PRM significantly improve the performance compared without response selection.
>
> >Q3:Part of the paper are confusing. For example, Athena-300K in line893 should be MC-300K? Athena-5K should be a subset of MC-300K?+9.9 in line354 should be 10.9?
>
> Thanks for pointing them out. In line893, Athena-300K should be MC-300K. Athena-5K is a (only part) subset of MC-300K. +9.9 should be +10.9 in line354. We have fixed the above typos in the revised reversion.

---

> ### Author Response · Authors · 2025-11-26
>
> Dear Reviewer KwTQ,
>
> We hope this message finds you well. As the discussion period is approaching its end (about **one week** remaining), we wanted to kindly check whether you have any additional comments or concerns that we could help clarify. We truly appreciate your time and effort in reviewing our work, and we are happy to provide any further details that might assist your evaluation.
>
> Thank you again for your valuable feedback.
>
> Best regards, Authors

---

> ### Author Response · Authors · 2025-11-27
>
> Dear Reviewer KwTQ,
>
> We hope this message finds you well. As the discussion period is approaching its end (with about one week remaining), we would like to kindly ask whether our responses have adequately addressed your concerns regarding our submission.
>
> We sincerely appreciate your valuable comments and the time you have taken to review our work. We would be very happy to provide further clarifications or engage in additional discussion if you have any remaining questions or suggestions.
>
> We believe that detailed and constructive discussion is highly beneficial to the research community, and we are grateful for your contributions to this process.
>
> Thank you again for your time and consideration.
>
> Best regards
>
> Authors

---

### Official Review · Reviewer_Pt8E · 2025-11-04

**Soundness:** 3
**Presentation:** 3
**Contribution:** 2
**Rating:** 4
**Confidence:** 3

**Summary:**

This work proposes Athena-PRM, a multimodal process reward model that assigns step-level rewards to reasoning trajectories. To generate high-quality process labels efficiently, it introduces a consistency filter that keeps only steps whose Monte Carlo hard labels agree between a weak and a strong model. Two training strategies are explored: initializing PRMs from an outcome reward model, and up-sampling negative step labels. Athena-PRM is evaluated in three scenarios: Best-of-N test-time scaling, direct step judgment, and reward-ranked fine-tuning. Across seven multimodal and two text-only benchmarks, Athena-PRM shows consistent gains.

**Strengths:**

- The paper presents a simple and effective label-quality filter via weak/strong completer consistency.
- This work achieves  strong multi-benchmark empirical performance. Three scenarios—BoN verification, direct step judgment, and RAFT—are specified and evaluated, demonstrating versatility.

**Weaknesses:**

- Limited validation scale and sensitivity of the consistency filter. The label accuracy validation uses only 50 queries , which is small and may not generalize; larger validation is not shown. Sensitivity to the number of MC samples T is not analyzed; all examples use T=8, yet hard labels depend heavily on T, affecting reliability.
- Misaligned RAFT baselines in “Evaluation on the Fine-Tuned Model” . The paper’s objective is to demonstrate Athena-PRM’s effectiveness, so RAFT should be compared against RAFT driven by alternative PRMs/ORMs (e.g., VisualPRM, text verifiers, ORMs), rather than against fine-tuned models with different training data and recipes.  The current setup conflates improvements from data selection, training pipeline, and model size with the reward model’s contribution, hindering clean attribution.
- Mathematical and notation clarity issues: 1) Step segmentation/labeling details (“\n\n”, “+/-” labels) lack precise token-level application description, which is important for reproducibility. 2)Minor typos, e.g., “Tabele 6” in Sec. 2.2; duplicated “ϕ2w” in Table 6 caption.
-  Reproducibility. No evidence of code, data, or model release to support reproducibility and result authenticity.

**Questions:**

Regarding Eq. (1) and Eq. (2), should there be a negative sign (i.e., standard negative log-likelihood)? What is r’s range and activation (e.g., sigmoid to ensure r∈(0,1)) to avoid log-domain issues for log r and log(1−r)?

---

> ### Author Response · Authors · 2025-11-19
> **Response to Pt8E - part (1/2)**
>
> We’d like to thank you for the comprehensive and constructive comments, and we now address each of the given concerns:
>
> >W1: Limited validation scale and sensitivity of the consistency filter. The label accuracy validation uses only 50 queries, which is small and may not generalize; larger validation is not shown. Sensitivity to the number of MC samples T is not analyzed; all examples use T=8, yet hard labels depend heavily on T, affecting reliability.
>
> Here we provide step accuracy of estimated process labels under different T and more examples. Due to the original PRM-800K containing >100K samples, running experiments on the full dataset requires too many GPU hours and we randomly choose 1000 samples.
>
> T| # samples |  weak completer    | strong completer    | consistency filter |
> :-----:|:------:|:-----:|:-----:|:-----:|
> 8| 50 | 78.2 | 83.4 | 94.1 |
> 8 | 1000  |78.0 | 82.8 | 91.0 |
> 16 | 50 | 78.4 | 81.8 |92.8 |
> 16 | 1000 | 78.6 | 81.8| 93.0 |
>
> Experimental results show that under different numbers of T and queries, using the consistency filter significantly improves accuracy of estimated process labels.
>
> >W2: Misaligned RAFT baselines in “Evaluation on the Fine-Tuned Model” . The paper’s objective is to demonstrateAthena-PRM’s effectiveness, so RAFT should be compared against RAFT driven by alternative PRMs/ORMs (e.g.,VisualPRM, text verifiers, ORMs), rather than against fine-tuned models with different training data and recipes.The current setup conflates improvements from data selection, training pipeline, and model size with the reward model’s contribution, hindering clean attribution.
>
> Because training data has ground truth answers, we do not compare with the baseline that using ORMs select data. In Tab.11, we compare with several baselines:
>
> | Model | Data | Athena-PRM | Accuracy |
> |:---|:---:|:---:|:---:|
> Qwen2.5-VL-7B | - | - | 68.1 |
> +RAFT | 30K| x | 68.7 |
> +RAFT | 30K |√| 69.8 |
> +RAFT|150K| √| 71.4 |
>
> For RAFT, we first generate 8 solutions using the current policy $\pi$ (here $\pi$ is Qwen2.5-VL-7B) for an input query x. Subsequently, we filter out solutions with incorrect answers and apply deduplication to remove highly similar responses.
> With 30K queries, we get 68.7% accuracy on MathVista without Athena-PRM and 69.8 accuracy with Athena-PRM to select responses. Furthermore, we increase data to 150K and get 71.4% accuracy on MathVista.

---

> ### Author Response · Authors · 2025-11-19
> **Response to Pt8E - part (2/2)**
>
> >W3: Mathematical and notation clarity issues: 1) Step segmentation/labeling details (“\n\n”, “+/-” labels) lack precise token-level application description, which is important for reproducibility. 2) Minor typos, e.g., “Tabele 6” in Sec.2.2; duplicated “ϕ2w” in Table 6 caption.
>
> We are sorry for confusing you. We explain related details as follows.
> (1) At test time, we prompt models to generate step-by-step solutions and each step is separated by `\n\n`. We split the solution using `\n\n` to get each step and we concatenate each step with **the special token** `<step>` to forward into Athena-PRM. After that, we could get the reward for each step with only one forward pass. We choose the probability of token `+` as the reward.
>
> `<step>` is added during the training stage as the special token. `+/-` denotes the correctness of each step. `+/-` has already existed in the vocabulary of LLMs. For example, in Qwen2.5-VL-7B, the token id of `+/-` is 10 and 12, respectively.
>
> We provide an example of python-style pseudo code to demonstrate this.
> For response generate by the policy model (e.g. Qwen2.5-VL-7B), the response is as follows:
> ```
> To determine the highest number…….
>
> 1. The black outer part of the watch a
>
> 2. The numbers on the bezel are xxxx
>
> xxxx
> ```
> We get the reward of each step as follows:
> ```
> response = """
> To determine the highest number…….
>
> 1. The black outer part of the watch a
>
> 2. The numbers on the bezel are ar
> """
> question = "xxxx"  # question for test
> response = """
> xxx
> """  # response generated by the policy model
>
> # load model and tokenizer
> model = xxx
> tokenizer = xxx
> good_token = "+" # token denotes correct steps
> bad_token = "-"  # token denotes incorrect steps
> step_token = "<step>"  # token for separate different steps
>
> step_tag_id = tokenizer.encode(step_token)[0]
> good_token_id =  tokenizer.encode(good_token)[0] # 10 in Qwen2.5-VL-7B
> bad_token_id =  tokenizer.encode(bad_token)[0] # 12 in Qwen2.5-VL-7B
>
> steps = response.split("\n\n")   # use \n\n to split steps in the response
>
> inputs = question + step_token.join(steps) + step_token # concatenate each step with special token <step>, e.g.
> inputs = tokenizer.encode(inputs)
> logits = model(inputs) # logits shape: [batch size, sequence length, vocabulary size]
> logits = logits[:,:,[good_token_id,bad_token_id]]  # only keep logits of +/-
> scores = logits.softmax(dim=-1)[:,:,0] # use probability of token + as the step reward
> step_scores = scores[input_id == step_tag_id]
> print(step_scores) # get the reward of each step
> ```
> (2) Thanks for pointing out, we have fixed the typo in the caption of Table 6 in our revised version.
>
> >W4: Reproducibility. No evidence of code, data, or model release to support reproducibility and result authenticity.
>
> We appreciate your feedback, and we believe in the importance of more transparent and open-source research for advancing scientific progress. The open-source plan is under discussion and internal check, and we are awaiting further guidance on this matter. In the submission, we provide as many details as possible about our method.
>
>
> >Q1: Regarding Eq. (1) and Eq. (2), should there be a negative sign (i.e., standard negative log-likelihood)? What is r’s rangeand activation (e.g., sigmoid to ensure r∈(0,1)) to avoid log-domain issues for log r and log(1−r)?
>
> Yes, it is a typo. There should be a negative sign in Eq.(1) and Eq.(2). We have fixed it in the revised version.
> We use the output probability at token `+` as the reward, so $r \in \left(0,1\right)$ to ensure the value of $r$ and $1-r$ lies within the domain of the log function.

---

> ### Author Response · Authors · 2025-11-26
>
> Dear Reviewer Pt8E,
>
> We hope this message finds you well. As the discussion period is approaching its end (about **one week** remaining), we wanted to kindly check whether you have any additional comments or concerns that we could help clarify. We truly appreciate your time and effort in reviewing our work, and we are happy to provide any further details that might assist your evaluation.
>
> Thank you again for your valuable feedback.
>
> Best regards, Authors

---

> ### Author Response · Authors · 2025-11-27
>
> Dear Reviewer Pt8E,
>
> We hope this message finds you well. As the discussion period is approaching its end (with about one week remaining), we would like to kindly ask whether our responses have adequately addressed your concerns regarding our submission.
>
> We sincerely appreciate your valuable comments and the time you have taken to review our work. We would be very happy to provide further clarifications or engage in additional discussion if you have any remaining questions or suggestions.
>
> We believe that detailed and constructive discussion is highly beneficial to the research community, and we are grateful for your contributions to this process.
>
> Thank you again for your time and consideration.
>
> Best regards
>
> Authors

---

### Author Response · Authors · 2025-11-19
**General response**

We sincerely thank all reviewers for their feedback and constructive suggestions. We particularly appreciate the acknowledgment of:
* simple and effective method (**Pt8E**, **gqQb**)
* strong multi-benchmark empirical performance and experiments (**Pt8E**,**KwTQ**, **gqQb**, **4HPA**)
* well-motivated (**KwTQ**)
* well-structured paper (**gqQb**)

We add new results including
* validation across different MC samples (T=8->16) on more queries (50->1000) (**Pt8E**)
* different backbones to train Athena-PRM (**KwTQ**, **gqQb**)
* scaling data from 5K to 60K (**4HPA**, **gqQb**)
* more results on other domains, including M3CoT and EMMA (**4HPA**)
* results of distillation (**4HPA**)

The summary of submission updates includes (we highlight all changes with red):
* add data scaling experiments in Sec.3.4 (**4HPA**, **gqQb**)
* fix minor typos. (**Pt8E**, **KwTQ**)

---

### Comment · Area_Chair_yTt6 · 2025-11-21
**Rebuttal Received - Next Steps**

Dear Reviewers,

The authors have submitted their rebuttal. Please review their responses and provide any follow-up, such as additional questions or revisions to your review.

As the initial scores varied significantly (8 to 4), I kindly encourage you to carefully consider the other reviewers' perspectives as you formulate your final recommendation.

Thank you for your contributions to this process.

Sincerely,
Your AC

---

### Author Response · Authors · 2025-11-28
**Does our response address your concerns?**

Dear **all reviewers**

We hope this message finds you well. As the discussion period is approaching its end (less one week remaining), we would like to kindly ask whether our responses have adequately addressed your concerns regarding our submission.

We sincerely appreciate your valuable comments and the time you have taken to review our work. We would be very happy to provide further clarifications or engage in additional discussion if you have any remaining questions or suggestions.

We believe that detailed and constructive discussion is highly beneficial to the research community, and we are grateful for your contributions to this process.

Thank you again for your time and consideration.

Best regards

Authors

---

### Meta-Review · Area_Chair_SUtt · 2026-01-03

**Summary:**

This paper proposes Athena-PRM, a multimodal PRM trained from pseudo step-level labels filtered via weak-strong completer consistency, with additional design choices (ORM initialization and negative upsampling). Reviewers generally agreed the idea is reasonable and the empirical results are strong on the chosen benchmarks. However, there are several concerns that affect confidence in the main claims: (i) the efficiency/"data-efficient" claim is not convincingly supported without a clear accounting of labeling cost and sample difficulty; (ii) multiple experiments (e.g., the RAFT section and some tables) involve gains from model/backbone differences, training recipe, and data selection, making attribution to the proposed PRM/data-filtering method unclear; (iii) there are insufficient ablation studies and comparisons to alternative data synthesis/selection strategies to ablate the contribution of the consistency filter; and (iv) reproducibility and clarity issues (e.g., notation, implementation details, and release commitments) remain a concern.

**Reviewer Concerns:**

### Concerns addressed by the rebuttal: ###

- One reviewer indicated that the additional experiments and clarifications in the rebuttal sufficiently addressed their main questions and maintained a positive assessment.

- The rebuttal added some new analyses and broader validation results, which partially responded to requests for more empirical evidence and scaling behavior.

### Outstanding concerns: ###

- The central claim of data efficiency remains weak. The paper does not provide a clear accounting of labeling cost, compute cost, or sample difficulty, making it hard to judge whether the approach is truly more efficient than alternatives.

- Several experimental comparisons are still difficult to interpret because gains may come from backbone choices, training recipes, or data differences rather than the proposed consistency-based filtering itself.

- The evaluation on certain benchmarks, including task selection and baselines, raises concerns about attribution and possible selection bias.

- Important ablations are missing, especially comparisons to simpler or alternative data selection strategies.

**Reviewer Scores:**

- Reviewer gqQb: original score 8, expected score 8. This reviewer stated that their concerns were addressed and would not change their rating.

- Reviewer Pt8E: original score 4, expected score 6. The added experiments and explanations likely improve confidence somewhat, but not enough to fully resolve the main concerns.

- Reviewer KwTQ: original score 4, expected score 4. Core issues around confounded comparisons and experimental design would likely keep the score unchanged.

- Reviewer 4HPA: original score 4, expected score 4. Despite additional results, concerns about efficiency claims, attribution, and clarity may remain.

---

### Decision · Program_Chairs · 2026-01-26

Reject